# New Solid Forms of Nitrofurantoin and 4-Aminopyridine Salt: Influence of Salt Hydration Level on Crystal Packing and Physicochemical Properties

**DOI:** 10.3390/molecules27248990

**Published:** 2022-12-16

**Authors:** Denis E. Boycov, Ksenia V. Drozd, Alex N. Manin, Andrei V. Churakov, German L. Perlovich

**Affiliations:** 1G.A. Krestov Institute of Solution Chemistry, Russian Academy of Sciences, 1 Akademicheskaya St., 153045 Ivanovo, Russia; 2Institute of General and Inorganic Chemistry, Russian Academy of Sciences, 31 Leninsky Prosp., 119991 Moscow, Russia

**Keywords:** nitrofurantoin, 4-aminopyridine, salt, hydrate formation, crystal structure, dehydration, stability, dissolution

## Abstract

The crystallization of the poorly soluble drug nitrofurantoin (NFT) with 4-aminopyridine (4AmPy) resulted in three multicomponent solid forms with different hydration levels: anhydrous salt [NFT+4AmPy] (1:1), salt monohydrate [NFT+4AmPy+H_2_O] (1:1:1), and salt tetrahydrate [NFT+4AmPy+H_2_O] (1:1:4). Each salt was selectively prepared by liquid-assisted grinding in the presence of acetonitrile or ethanol/water mixture at a specific composition. The NFT hydrated salts were characterized using single crystal X-ray diffraction. The [NFT+4AmPy+H_2_O] salt (1:1:1) crystallized as an isolated site hydrate, while the [NFT+4AmPy+H_2_O] salt (1:1:4) crystallized as a channel hydrate. The dehydration processes of the NFT salt hydrates were investigated using differential scanning calorimetry and thermogravimetric analysis. A powder dissolution experiment was carried out for all NFT multicomponent solid forms in pH 7.4 phosphate buffer solution at 37 °C.

## 1. Introduction

One of the important aspects of drug development is to determine which solid form of the active pharmaceutical ingredient (API) will be the most preferable for its scale-up [1]. It is well-known that APIs can exist in various solid forms (polymorphs, hydrates/solvates, salts, cocrystals) and their physicochemical properties can differ strikingly from the properties of the parent drug compound [2,3,4,5,6,7,8,9,10]. Therefore, it is important to identify and characterize all possible solid forms of APIs in the early stages of drug development in order to use the optimal form of API with the best characteristics in the pharmaceutical formulation [1].

Poor water solubility is one of the major challenges for many APIs, which in turn is the reason for their poor bioavailability [11]. The preparation of multicomponent solid forms (salts, cocrystals) based on poorly soluble APIs is one of the promising solutions to this problem. This approach allows the modification of a number of physicochemical properties of the API (solubility, dissolution rate, melting point, permeability, physical stability, tabletability, etc.) without changing its inherent pharmacological properties [12]. Some studies have shown that the solubility of poorly soluble APIs can be increased hundreds or even thousands of times by salt or cocrystal formation [13,14,15,16]. However, despite the many advantages of cocrystallization, only a very limited number of pharmaceutical cocrystals have been approved and marketed as drug products to date, such as Suglat^®^, Entresto^®^, Steglatro^®^, Depakote^®^, Lexapro^®^, and Betachlor^®^ [17]. Unlike cocrystals, which are just beginning to enter the pharmaceutical market, more than 50% of drugs are sold as organic salts [18].

As well as poor water solubility, approximately one third of pharmaceutical solids can form hydrates during the manufacturing process [19]. However, the formation of multicomponent solid forms does not prevent the hydration of APIs. The formation of hydrates for multicomponent solids, as well as their stability, is still an unpredictable factor. Moreover, the tendency for hydrate formation is especially high for salt forms, which could be due to the strong interaction potential of water with ions in the salt structure [20,21]. Additionally, the imbalance between the number of hydrogen bond donors and acceptors in the compounds can also contribute to the formation of hydrates [22,23,24]. Water, in comparison with other organic solvents, due to the small size of its molecules, is able to integrate into the voids of the crystal structure, thus forming channel hydrates. On the other hand, water can bind salt components into a stable crystal structure, acting as both hydrogen bond donor and acceptor [25,26]. The incorporation of water molecules in the crystal structure will affect many of the physicochemical properties of the API or its multicomponent solid compared to the anhydrous form. For example, hydrated forms may exhibit better compressibility and tabletability compared to anhydrous forms [27,28]. In such cases, it is advisable to give preference to hydrated forms of the API or pharmaceutical salt to prevent form change during manufacture or storage. On the other hand, hydrated forms often have lower solubility than anhydrous forms [29,30,31]. In this regard, the choice between hydrated and anhydrous forms can be a decisive factor in drug manufacturing.

In this work, nitrofurantoin (NFT) was chosen as the object of study (Figure 1). NFT is an antibiotic used to treat and prevent lower urinary tract infections, such as cystitis [32]. NFT shows polymorphism and exists in two forms (α and β). The β-polymorph is the commercial form of nitrofurantoin [33]. In addition to the anhydrous polymorphs, there are two known monohydrate forms of NFT [34]. As we have previously demonstrated, the stability of the NFT anhydrous form in water is extremely low. When dissolved in water, NFT almost immediately transforms into a more thermodynamically stable hydrated form, which is accompanied by a sharp decrease in API solubility [29,35]. Preparation of multicomponent crystals with highly soluble coformers is one of the most promising approaches to increase the solubility and dissolution rate of NFT [36,37,38,39,40,41]. However, the hydration of NFT cocrystals and salts also occurs, for example, with L-arginine [40], melamine, 4,4′-bipyridine, 1,2-bis(4-pyridyl ethane) [41], and 4-aminopyridine [37]. The tendency for hydration of both the parent NFT and its multicomponent crystals can be specifically associated with the imbalance between the number of hydrogen bond donors and acceptors in the API molecule: one donor against six acceptors.

The aim of the present study was to study hydrate formation of the NFT salt with 4-aminopyridine (Figure 1). 4-Aminopyridine (4AmPy) is a potassium channel blocker used to treat various neurological disorders [42]. Moreover, there is a literature review describing the efficacy of this drug compound in the treatment of lower urinary tract symptoms in a patient with idiopathic nystagmus [43]. The monohydrate salt of NFT and 4AmPy was first obtained by Segalina et al. [37]; however, the physicochemical properties of the multicomponent solid were not investigated due to problems associated with obtaining a pure phase. In this paper, we present the selective synthesis of not only the known form, but also two new forms, one of which is the anhydrous salt [NFT+4AmPy] (1:1) and the second is the salt tetrahydrate [NFT+4AmPy+H_2_O] (1:1:4). All three multicomponent solids were obtained both by mechanochemical and crystallization methods and characterized by single crystal and/or powder X-ray diffraction. Differential scanning calorimetry and thermogravimetric experiments were carried out to study the thermophysical characteristics of each salt form. Finally, powder dissolution experiments were carried out in pH 7.4 phosphate buffer solution at 37 °C to compare the dissolution profiles of the parent NFT and its multicomponent crystals. The conducted studies allowed us to conclude how the packing of water molecules in the crystal structures of the NFT and 4AmPy salts with different hydration levels affected the physicochemical properties.

## 2. Results and Discussion

### 2.1. Crystal Structure Analysis

[NFT+4AmPy+H_2_O] salt (1:1:1) crystallizes in the monoclinic space group P2_1_/n, and the asymmetric unit contains one ion each of NFT and 4AmPy and one water molecule (Appendix A). Herein, NFT adopts a “twisted” conformation (conformer II [44]), where the nitrofuran fragment is rotated by 180° around the C–C bond in relation to the rest of the molecule. Previously, we reported that this conformation is not specific to the NFT molecule since it occurs only in 15% of all determined crystal structures for this API [38]. In the crystal structure, NFT and 4AmPy ions are linked by N12^+^–H1···N1^−^ interaction. The adjacent NFT-4AmPy dimers are connected to each other via N11–H11···O1 hydrogen bond to form a twisted zigzag chain (Figure 2a). The 3D crystal packing of the salt is fulfilled by the water molecule. The water molecules act as a bridge, which connects four independent chains of NFT and 4AmPy via strong O21–H20···O2 and O21–H21···O2 hydrogen bonds to generate a tetrameric ring motif (Figure 2b). Based on the location of the water molecules, the [NFT+4AmPy+H_2_O] salt (1:1:1) is a class of hydrate termed an isolated site hydrate, where the water molecules are found within discrete pockets isolated from direct contact with other water molecules [21].

[NFT+4AmPy+H_2_O] salt (1:1:4) crystallizes in the orthorhombic space group Pnma, and the asymmetric unit contains one ion each of NFT and 4AmPy and four water molecules (Appendix A). Notably, NFT adopts a frequently observed conformation (conformer I [44]), in contrast to the [NFT+4AmPy+H_2_O] salt (1:1:1). The crystal structure consists of NFT-4AmPy dimers bonded via N11^+^—H1···N1^−^ interaction. The hydrogen bond distances suggest that a stronger pyridinium-imide heterodimer is observed here than in the salt monohydrate (Appendix A). These dimers are further developed into a zigzag chain (Figure 3a). The resulting chain, in contrast to the chain in the NFT salt monohydrate, is practically planar. This is due to a different NFT conformation. Thus, the neighboring NFT-4AmPy dimers are connected via bifurcated N12—H10···O3 and N12—H11···O1 hydrogen bonds. The water molecules form tetramers that are hydrogen-bonded to the neighboring NFT ions via O21—H21···O1 and O23—H24···O2 hydrogen bonds (Figure 3a). The chains are assembled into 2D layers via weak C—H···O interactions and further stacked into a 3D channel structure with water molecules sitting in cavities along the b-axis (Figure 3b). Therefore, [NFT+4AmPy+H_2_O] salt (1:1:4) is a class of hydrate termed a channel hydrate. The Hydrate Analyser tool implemented in Mercury software was used to determine the water interaction map. The water volume occupied in the unit cell is 441.42 Å^3^, which is 24.5% of the total unit cell volume (Appendix A).

### 2.2. Evaluation of the Formation Pathways of the NFT Salts in LAG

Due to the fact that the single crystals of the [NFT+4AmPy+H_2_O] (1:1:1) and [NFT+4AmPy+H_2_O] (1:1:4) salts were obtained by crystallization from ethanol (96%) or ethanol/water mixture (50:50 *v*/*v*), respectively, it can be concluded that the water content of organic solvent is the main variable influencing the formation of various hydrated forms of the NFT salt. Previously, Segalina et al. reported that obtaining a pure form of NFT salt monohydrate by grinding is not a trivial task [37]. We assume that the difficulty in obtaining the salt monohydrate may be due to its partial transition to the tetrahydrate form during the mechanochemical reaction, which was not previously known. We previously observed a similar sensitivity of salts with different hydration levels to the water content of the water-organic mixture for norfloxacin salts with fumaric acid [45]. In this regard, we tried to determine the influence of the ethanol/water composition on the ability to obtain each hydrated form of the NFT salt as a result of a mechanochemical reaction. A series of grinding experiments was conducted with the physical mixture (NFT+4AmPy) in a 1:1 molar ratio in the presence of ethanol/water mixtures with various compositions. The experimental PXRD patterns of the obtained powder samples were compared with the PXRD patterns calculated from SCXRD data for the NFT hydrated salts (Figure 4).

The results indicated that the percentage of water in the ethanol/water solvent mixture should not exceed 9% to obtain a pure form of the NFT salt monohydrate by LAG. The increasing water content in the cosolvent mixture resulted in the appearance of characteristic peaks of the NFT salt tetrahydrate on the PXRD pattern (peaks at 2θ: 7.6° and 15.2°). The preponderance of the tetrahydrate form compared with the monohydrate form of the NFT salt in the powder sample occurred only when the ethanol percentage in the ethanol/water mixture was reduced to almost 50%. To obtain the [NFT+4AmPy+H_2_O] salt (1:1:4) without residual traces of the NFT salt monohydrate via LAG, the ethanol content should not exceed 30%. It should be noted that the pure form of the NFT salt tetrahydrate was obtained by grinding with additional water as the solvent. Moreover, the grinding of the NFT salt monohydrate with additional water resulted in the transformation of the initial phase into the tetrahydrate form (Appendix A). During the grinding experiments, it was noted that the powdered samples of the NFT salts with different hydration levels were characterized by various shades of yellow: the powder of the salt monohydrate had a pale yellow color while the powder of the salt tetrahydrate had a more saturated shade of yellow (Appendix A).

In addition to the two hydrated forms of the NFT salt, a new phase identified as the anhydrous NFT salt [NFT+4AmPy] (1:1) was obtained by crystallization and grinding in the presence of acetonitrile. Interestingly, acetonitrile is an organic solvent which is most frequently used to produce single crystals of the anhydrous/unsolvated NFT multicomponent solids [38,40,41,44,46,47]. Despite numerous attempts, we failed to grow single crystals of the [NFT+4AmPy] salt (1:1) suitable for SCXRD. However, the resulting powdered sample of the new phase obtained by crystallization from acetonitrile of NFT and 4AmPy in a 1:10 molar ratio was analyzed by PXRD (Figure 5).

The new phase was characterized by a unique set of reflections, which differed from both the parent components and the two hydrated forms of the salt. The experimental PXRD pattern of the sample obtained by grinding of the physical mixture in a 1:1 molar ratio in the presence of acetonitrile completely coincided with the PXRD of the anhydrous phase obtained by crystallization. The absence of the solvent used and/or water traces in the powder sample of the [NFT+4AmPy] salt (1:1) was confirmed by DSC/TG. The grinding of the [NFT+4AmPy] salt (1:1) in the presence of water resulted in the transformation of the initial phase into the tetrahydrate form (Appendix A). The results of the mechanochemical experiments are depicted in Figure 1. It was found that regardless of the initial phase (physical mixture, [NFT+4AmPy] (1:1) or [NFT+4AmPy+H_2_O] (1:1:1)), the grinding of the powdered sample in the presence of water led to the formation of the [NFT+4AmPy+H_2_O] salt (1:1:4).

### 2.3. Thermal Analysis

The tehermal behavior of [NFT+4AmPy] (1:1), [NFT+4AmPy+H_2_O] (1:1:1), [NFT+4AmPy+H_2_O] (1:1:4) was studied by DSC and TGA. The DSC and TG curves for all multicomponent forms of NFT are shown in Figure 6. Thermophysical data for the hydrated forms of NFT salt are summarized in Table 1.

The DSC and TG curves obtained for the [NFT+4AmPy] salt (1:1) confirmed that this phase was an anhydrous/unsolvated form of the NFT and 4AmPy salt. The DSC curve revealed a single endothermic peak with T_onset_ = 122.9 ± 0.2 °C (∆H_fus_ = 57.2 ± 0.5 J·g^−1^), indicating the melting of salt. An exothermic event immediately after the endothermic peak was associated with sample decomposition at temperatures above 140 °C. According to TG analyses, both of these effects were accompanied by a significant weight loss of the sample (~30%).

Thermal analysis of the NFT salt hydrates revealed the following results. A single endothermic event observed in the DSC curve of both samples corresponded to the dehydration process with the onset temperature of 99.3 ± 0.2 °C for the NFT salt monohydrate or 38.4 ± 0.2 °C for the NFT salt tetrahydrate. The dehydration was followed by a weight loss event. The TG curves of the [NFT+4AmPy+H_2_O] salt (1:1:1) and the [NFT+4AmPy+H_2_O] salt (1:1:4) showed a weight loss of 5.09% or 17.76%, respectively, which was consistent with the calculated weight loss (5.14% or 17.83%, respectively). This was consistent with the release of one or four water molecules from the corresponding NFT salts. After dehydration, the endothermic peaks of the two NFT salts corresponding to the melting were not observed. This indicated that the release of water led to destruction of the crystal structure of the salts and, as a result, amorphization because the components failed to rearrange and pack immediately. A similar thermal behavior was also observed for the NFT solvate with 3-picoline and its hydrated form [44]. The large difference between the dehydration temperatures of the NFT salt hydrates can be related to the bonding environment features in the crystal structures and the binding strength of the solvent molecules. The binding strength of the water molecules in the NFT hydrated salts can be estimated by calculating the vaporization enthalpy (Δ*H_s_*) of the salt-bound solvent using the following equation [48]:(1)$$\Delta H_{s}=(\Delta H_{desolv}^{T}\times 100/\Delta m_{s})\times M_{s}$$
where $\Delta H_{desolv}^{T}$ is the enthalpy of desolvation/dehydration determined from the DSC data, Δ*m_s_* is the mass loss percent measured in the TG experiment, and *M_s_* is the solvent molecular weight. The resulting Δ*H_s_* values for the NFT salt hydrates are given in Table 1.

In the [NFT+4AmPy+H_2_O] salt (1:1:1), the Δ*H_s_* value was 12% higher than in the pure liquid (vaporization enthalpy of water ≈ 40.7 kJ·mol^−1^). This indicated stronger interactions of water molecules with salt components than in pure water and the NFT salt tetrahydrate. Indeed, in the crystal structure of the NFT salt monohydrate, the O—H···O hydrogen bonds formed between NFT ions and water molecules were the shortest hydrogen bonds among all the interactions in the crystal structure. Moreover, the water molecules played an essential role in the stabilization of the 3D structure of the [NFT+4AmPy+H_2_O] salt (1:1:1) linking the chains of the NFT-4AmPy dimers (Figure 2).

In the [NFT+4AmPy+H_2_O] salt (1:1:4), the Δ*H_s_* value was 24% lower than in pure water. The water molecules in the NFT salt tetrahydrate are located in the cavities; therefore, they can freely leave the crystal structure during heating. To confirm this hypothesis, a powdered sample of the [NFT+4AmPy+H_2_O] salt (1:1:4) was heated to 100 °C, the temperature at which dehydration ends, and cooled to 20 °C (Appendix A). The sample obtained after the first heating cycle was analyzed by PXRD (Appendix A). It was found that dehydration led to a significant decrease in sample crystallinity, which was proven by the intensity reduction of the peaks. However, the characteristic peaks on the PXRD pattern of the sample obtained after dehydration coincided with the peaks of the [NFT+4AmPy+H_2_O] salt (1:1:4).

### 2.4. Powder Dissolution

It is known that nitrofurantoin is a drug compound with poor solubility. Moreover, NFT spontaneously transforms to a monohydrate form in aqueous medium, the solubility of which is lower than that of the anhydrous form [29]. Therefore, it was interesting to determine whether the solubility of NFT would increase as a result of its salt formation with 4AmPy and to investigate the behavior of various salt forms during their dissolution. Powder dissolution experiments were carried out in pH 7.4 phosphate buffer solution at 37 °C, and the resulting dissolution profiles are presented in Figure 7.

Based on the obtained profiles, it was found that the dissolution of all NFT multicomponent solid forms was characterized by the well-known “spring and parachute” pattern. Such behavior indicated that the NFT salts, regardless of their hydration level, underwent a phase transformation during dissolution. This was also confirmed by the PXRD analysis of the residual materials collected at the end of the experiment (Appendix A). In all cases, it was found that NFT multicomponent solids transformed to a less soluble hydrated phase of the API.

The [NFT+4AmPy] (1:1) and [NFT+4AmPy+H_2_O] (1:1:4) salts had similar dissolution profiles. They were characterized by a faster dissolution rate than that of the parent NFT and NFT salt monohydrate. The maximum concentrations (C_max_) of both of the NFT salts reached in the first 15–20 min of the dissolution experiment were the same (2.60 × 10^−3^ M) and were 24% higher than the NFT monohydrate solubility (2.11 × 10^−3^ M). However, an initial rapid release of NFT from the salt forms was followed by a sharp decrease in the API concentration. After dissolution of the [NFT+4AmPy] (1:1) and [NFT+4AmPy+H_2_O] (1:1:4) salts for 30 min, the NFT concentration decreased to a similar plateau concentration as that of the NFT monohydrate. The dissolution behavior of the anhydrous and hydrated salts confirmed that the water molecules in the crystal structure of the [NFT+4AmPy+H_2_O] salt (1:1:4) were weakly hydrogen-bonded to the salt components, and salt hydration did not significantly contribute to NFT solubility and stability.

The [NFT+4AmPy+H_2_O] salt (1:1:1) dissolution was characterized by a gradual increase in the NFT concentration in the first 90 min of the experiment. The C_max_ value of the NFT salt monohydrate (2.94 × 10^−3^ M) was 38% higher than the NFT monohydrate solubility. Furthermore, due to the NFT concentration decreasing more slowly compared to the other salt forms, the API supersaturation level in the buffer solution was maintained for almost 6 h. The prolonged stability of the [NFT+4AmPy+H_2_O] salt (1:1:1) during the dissolution experiments was associated precisely with the specific crystal packing of the water molecules in the NFT salt monohydrate. Thus, taking into account the poor water solubility and stability of the parent NFT, the results presented in this paper highlight the potential application of [NFT+4AmPy+H_2_O] salt (1:1:1) in the development of new nitrofurantoin formulations.

## 3. Experimental Section

### 3.1. Materials

Nitrofurantoin was purchased from Acros Organics (Geel, Belgium) and its PXRD pattern was found to correspond to the β-polymorph form of this drug with Cambridge Structural Database (CSD) code LABJON02. 4-Amynopyridine was obtained from Sigma-Aldrich (St. Louis, MO, USA). All of the chemicals were used as received without further purification. Chromatographic or analytical grade solvents were used in the experiments.

### 3.2. Sample Preparation

#### 3.2.1. Solution Crystallization

Single crystals of the NFT hydrated salts ([NFT+4AmPy+H_2_O] (1:1:1) and [NFT+4AmPy+H_2_O] (1:1:4)) were successfully obtained by the slow evaporation method. The orange needle crystals of [NFT+4AmPy+H_2_O] (1:1:1) were prepared by dissolving a physical mixture of NFT (10.1 mg) and 4AmPy (39.9 mg) in a 1:10 molar ratio in ethanol 96% (5 mL). The orange prism crystals of [NFT+4AmPy+H_2_O] (1:1:4) were prepared by dissolving a physical mixture of NFT (10.1 mg) and 4AmPy (39.9 mg) in a 1:10 molar ratio in a mixture of ethanol/water (50:50 *v*/*v*), in a sufficient quantity to ensure full dissolution. The crystallizing dishes containing the resulting solutions were covered by parafilm perforated with a few small holes, and the solutions allowed to evaporate slowly in the dark at room temperature. Single crystals suitable for SCXRD were obtained after a few days. Despite numerous attempts, we were unable to obtain good quality single crystals for the anhydrous NFT salt [NFT+4AmPy] (1:1). The powdered sample of [NFT+4AmPy] (1:1) was prepared by dissolving a physical mixture of NFT (10.1 mg) and 4AmPy (39.9 mg) in a 1:10 molar ratio in acetonitrile. The resulting powder was isolated and characterized by PXRD and DSC/TG techniques.

#### 3.2.2. Liquid-Assisted Grinding (LAG)

The grinding experiments were performed using a Fritsch planetary micro mill (Model Pulverisette 7; Fritsch, Idar-Oberstein, Germany) in 12 mL agate grinding jars with ten 5 mm agate balls at a rate of 500 rpm for 60 min. In a typical experiment, 50 mg of a physical mixture of NFT and 4AmPy in a 1:1 molar ratio were placed into a grinding jar and 50 μL of solvent or solvent mixture was added with a micropipette. The phase composition of the resulting bulk samples was analyzed by PXRD.

### 3.3. X-ray Diffraction

#### 3.3.1. Single Crystal X-ray Diffraction (SCXRD)

The X-ray diffraction data for the [NFT+4AmPy+H_2_O] (1:1:1) single crystals were collected on a Bruker D8 Venture machine. Additionally, the [NFT+4AmPy+H_2_O] (1:1:4) single crystals were analyzed on a Bruker SMART APEX II diffractometer with graphite-monochromated Mo-Kα radiation (λ = 0.71073 Å). Adsorption corrections based on measurements of equivalent reflections were applied [49]. The structures were solved by direct methods and refined by full-matrix least-squares on F^2^ with anisotropic thermal parameters for all the non-hydrogen atoms [50]. All hydrogen atoms were found from different Fourier maps and refined isotropically. In the [NFT+4AmPy+H_2_O] salt (1:1:4), the hydrogen atoms of the water molecules are disordered and they were refined with restraints. The crystal data, data collection, and details of structure refinement are summarized in Table 2. Relevant hydrogen bonding interactions and their geometries are listed in Appendix A. The crystallographic data were deposited with the Cambridge Crystallographic Data Centre as supplementary publications under CCDC numbers 2216150 and 2216151. This information can be obtained free of charge from the Cambridge Crystallographic Data Centre at www.ccdc.cam.ac.uk/data_request/cif (accessed on 28 October 2022).

#### 3.3.2. Powder X-ray Diffraction (PXRD)

PXRD analysis of the NFT samples was performed on a D2 PHASER diffractometer (Bruker AXS, Karlsruhe, Germany) with Cu-Kα radiation (λ = 1.54187 Å) at 30 kV and 10 mA, equipped with a Lynxeye XE-T high-resolution position sensitive detector. The PXRD patterns were recorded over the range of 5–30° (2θ) with a step size of 0.02° and dwell time of 1 s.

### 3.4. Thermal Analyses

#### 3.4.1. Differential Scanning Calorimetry (DSC)

The thermal analysis was carried out using a differential scanning calorimeter equipped with a refrigerated cooling system (Perkin Elmer DSC 4000, Waltham, MA, USA). The sample was heated in a sealed aluminum sample holder at a rate of 10 °C min^−1^ under a nitrogen atmosphere. The unit was calibrated with indium and zinc standards. The accuracy of the weighing procedure was ±0.01 mg.

#### 3.4.2. Thermogravimetric Analysis (TGA)

TGA was performed using a TG 209 F1 Iris thermomicrobalance (Netzsch, Selb, Germany). Approximately 10 mg of the bulk sample was added to a platinum crucible. The samples were heated at a constant heating rate of 10 °C min^−1^ and purged throughout the experiment under a dry argon stream at 30 mL min^−1^.

### 3.5. Dissolution Studies

The powder dissolution experiments were carried out using a USP-certified Electrolab EDT-08LX dissolution tester (Navi Mumbai, India) applying the USP II paddle method for 360 min. Approximately 215 mg of pure NFT or an NFT-equivalent amount of the salt (non-sink conditions) was added to 300 mL of pH 7.4 buffer solution with a paddle speed of 100 rpm at 37.0 ± 0.1 °C. Aliquots of 1 mL with medium reposition were obtained by syringe at predetermined time intervals (5, 10, 15, 20, 30, 45, 60, 90, 120, 180, 240, 300, and 360 min). The samples were filtered using a Rotilabo PTEF syringe filter (Tullagreen, Ireland) with 0.2 μm pores. The concentrations of NFT and 4AmPy in the solution phase were determined after suitable dilution by HPLC. The results are stated as the average of three replicated experiments. The solution pH was measured at the beginning and end of each dissolution experiment. After the dissolution experiments, the solid residues were collected and dried at room temperature for PXRD analysis.

### 3.6. High-Performance Liquid Chromatography (HPLC)

HPLC was performed on an LC-20 AD Shimadzu Prominence model (Tokyo, Japan) equipped with a PDA detector and Luna C-18 column (150 mm × 4.6 mm i.d., 5 μm particle size, and 100 Å pore size). The column temperature was set to 40 °C. Elution of the samples was achieved using a mobile phase consisting of acetonitrile and 0.1% aqueous solution of trifluoroacetic acid mixed in a 15:85 (*v*/*v*) ratio using an isocratic regime at a flow rate of 1 mL·min^−1^. The injection volume was 20 μL. UV detection of NFT and 4AmPy was carried out at wavelengths of 265 and 262 nm, respectively.

## 4. Conclusions

Three multicomponent solid forms of nitrofurantoin with 4-aminopyridine (anhydrous salt, salt monohydrate, and salt tetrahydrate) were obtained and studied. Each salt was selectivity obtained by mechanochemical reaction in the presence of acetonitrile or ethanol/water mixtures of particular compositions. The crystal structures of the nitrofurantoin hydrated salts were determined and analyzed. The crystal packing analysis suggested that the hydrated salts crystallized into distinct crystal structures, which had a direct influence on differences in the physicochemical properties of the solid forms. The nitrofurantoin salt monohydrate, termed an isolated site hydrate, showed greater thermal stability than the salt tetrahydrate containing water molecules in channels. This was confirmed by higher values of dehydration temperature and vaporization enthalpy for the salt monohydrate compared to those of the salt tetrahydrate. Dissolution studies of the nitrofurantoin multicomponent solids in pH 7.4 buffer solution indicated that the anhydrous salt and its tetrahydrate form had higher dissolution rates than the parent drug and the salt monohydrate. In turn, the nitrofurantoin salt monohydrate showed the highest maximum concentration of the drug compound maintained over a prolonged time period.

## Data Availability

The results obtained for all experiments performed are shown in the manuscript and SI, the raw data will be provided upon request.

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
