# Peer review of "New Solid Forms of Nitrofurantoin and 4-Aminopyridine Salt: Influence of Salt Hydration Level on Crystal Packing and Physicochemical Properties"

_molecules, 2022, doi:10.3390/molecules27248990_

Round 1
Reviewer 1 Report
Title: New Solid Forms of Nitrofurantoin and 4-Aminopyridine Salt: Influence of Salt Hydration Level on Crystal Packing and Physicochemical Properties
Synopsis: In this manuscript the authors describe the synthesis and characterization of three new multicomponent solid forms of nitrofurantoin and 4-aminopyridine salts. The experiments carried out are consistent and they are adequate with the work (synthesis, crystal structure analysis, evaluation of the formation pathways of the NFT salts in LAG, thermal analysis and dissolution studies).
Critique: In general, the manuscript has the parameters to be published in this journal; the manuscript is very clear and concise, the references are good, the English is very satisfactory. This is a very good of work, but the authors need to address some of the typographical suggestion listed below:
Line 61: Change “anhydrate form” to “anhydrous form”?
Line 217: In ΔHfus, “fus” in lower case.
Reference 6: Crystal Growth & Design 2022, 22, (11), 6703-6716. doi: 449 10.1021/acs.cgd.2c00883.450
Reference 17: Frontiers in Pharmacology 2021, 12, 780582.
Reference 21: Journal of Drugs and the Pharmaceutical Sciences.
Reference 23: European Pharmaceutical Review 2014, 5 (28483October 2014).:‹ https://www.europeanpharmaceuticalreview.(remove this space)com/article/27753/pharmaceutical-salts-small-molecule-drugs/›, 484 cited 2021, 19.485
Referencia 32: Australian Prescriber.
Referencia 35: Chemical & Pharmaceutical Bulletin. (upper and lower case)
Reference 43: Interntaional Neurourology Journal. Full name of the journal.
Author Response
Response to Reviewer 1 Comments:
Title: New Solid Forms of Nitrofurantoin and 4-Aminopyridine Salt: Influence of Salt Hydration Level on Crystal Packing and Physicochemical Properties
Synopsis: In this manuscript the authors describe the synthesis and characterization of three new multicomponent solid forms of nitrofurantoin and 4-aminopyridine salts. The experiments carried out are consistent and they are adequate with the work (synthesis, crystal structure analysis, evaluation of the formation pathways of the NFT salts in LAG, thermal analysis and dissolution studies).
Critique: In general, the manuscript has the parameters to be published in this journal; the manuscript is very clear and concise, the references are good, the English is very satisfactory. This is a very good of work, but the authors need to address some of the typographical suggestion listed below:
- Line 61: Change “anhydrate form” to “anhydrous form”?
- Line 217: In ΔHfus, “fus” in lower case.
- Reference 6: Crystal Growth & Design 2022, 22, (11), 6703-6716. doi: 449 10.1021/acs.cgd.2c00883.450
- Reference 17: Frontiers in Pharmacology2021, 12, 780582.
- Reference 21: Journal ofDrugs and the Pharmaceutical Sciences.
- Reference 23: European Pharmaceutical Review2014, 5 (28483October 2014).:‹ https://www.europeanpharmaceuticalreview.(remove this space)com/article/27753/pharmaceutical-salts-small-molecule-drugs/›, 484 cited 2021, 19.485
- Referencia 32: Australian P
- Referencia 35: Chemical& Pharmaceutical Bulletin. (upper and lower case)
- Reference 43: InterntaionalNeurourology Journal. Full name of the journal.
Response:
The authors are grateful for the Reviewer 1 comments that improved the quality of the manuscript. The suggestions were included in the new version of the manuscript and are highlighted in red color.
Reviewer 2 Report
This is an excellent paper and I recommend its publication with just minor points. The Space Group should be written P21/n not P21/n. A 21 screw axis is present in SG 14.
In the 1:1:4 compound the water molecules form a tetramer however in the tables and pictures they contain three not four molecules O21, O23 and O25. Could this point be made clearer.
Author Response
Response to Reviewer 2 Comments:
Comment 1:
The Space Group should be written P21/n not P21/n. A 21 screw axis is present in SG 14.
Response:
Thanks for your comment. The text has been corrected and highlighted in red color.
Comment 2:
In the 1:1:4 compound the water molecules form a tetramer however in the tables and pictures they contain three not four molecules O21, O23 and O25. Could this point be made clearer.
Response:
As it was mentioned in the experimental section, hydrogen atoms of water molecules are disordered. Moreover, disorder cannot be eliminated inherently. It is a basic property of a give crystal and is caused by the presence of a mirror plane of symmetry in it. It was found that both organic fragments (NFT and 4AmPy) lie in the crystallographic plane m with occupation 0.5 per each. Two oxygen atoms (O21 and O23) are located in the same plane, and their occupancy is also 0.5. One of their hydrogen atoms also lies in this plane, while the other two occupy general positions and, therefore, are not strictly ordered with respect to the same plane with occupancies of 0.5. The oxygen atom O25 occupies a general position and its occupancy is 1. Both of its hydrogen atoms occupy general positions. However, they are forcedly not ordered over two positions due to the disorder of the hydrogen atoms at the O21 and O23 atoms, which form hydrogen bonds with O25. Thus, the occupancy ratio of organic fragments and water is 0.5:0.5:(0.5+0.5+1) or 1:1:4.
We decided not to describe in detail the disordering of water molecules both in the experimental and in Results and discussion sections, since this is not a unique case and does not affect the main conclusions of the work. It seems to us that the .cif file attached to the article and a brief mention in the experimental section is enough.
Reviewer 3 Report
The authors report the formation of new multicomponent salt forms of the drug Nitrofurantoin with 4-Aminopyridine. The article offers insights into the crystal structure of the new forms while also discussing their thermal properties and dissolution. Overall, the manuscript is very well written and is easy to follow. The results and figures are presented clearly, and I congratulate the authors for the work.

Author Response
Response to Reviewer 3 Comments:
The authors report the formation of new multicomponent salt forms of the drug Nitrofurantoin with 4-Aminopyridine. The article offers insights into the crystal structure of the new forms while also discussing their thermal properties and dissolution. Overall, the manuscript is very well written and is easy to follow. The results and figures are presented clearly, and I congratulate the authors on this work. I only have minor edits and suggestions for the authors to consider.
Comment 1:
Page 2 line 72: Recommend changing “monohydrate polymorphs of NFT” to “monohydrate forms of NFT”.
Response:
The text has been corrected and highlighted in red color.
Comment 2:
Page 6 line 183: I am curious about the solvent selection during crystallization of the anhydrous NFT salt. Was acetonitrile selected based on prior knowledge or were other solvents screened as well?
Response:
The solvent (acetonitrile) used for crystallization of the anhydrous NFT salt with 4AmPy was selected based on our prior knowledge. The fact is that the present paper is a continuation of previously conducted study of the nitrofurantoin multicomponent crystals (Surov A.O., Voronin A.P., Drozd K.V., Volkova T.V., Vasilev N., Batov D., Churakov A.V., Perlovich G.L. Extending the range of nitrofurantoin solid forms: effect of molecular and crystal structure on formation thermodynamics and physicochemical properties. Cryst. Growth Des., 2022, 22(4), 2569-2586). It was found that acetonitrile is an optimal organic solvent for preparation of single crystals of the anhydrous NFT multicomponent crystals with a number of benzamide derivatives. Besides, it is known that nitrofurantoin is poorly soluble in most organic solvents. Therefore, it was decided to use primarily acetonitrile for crystallization of the anhydrous salt.
Comment 3:
Page 6 line 187: is the ratio supposed to be 1:1 rather than 1:10?
Response:
The molar ratio (1:10) of the components used for crystallization of anhydrous salt is correct. As mentioned in the experimental section, single crystals for both NFT hydrated salt were also prepared by dissolving the physical mixture in a 1:10 molar ratio in ethanol or ethanol/water mixture. The use of an excess of 4AmPy in the solution crystallization experiments was associated with a large difference in solubility of parent compounds in acetonitrile or ethanol.
Comment 4:
Page 9 line 258: Should be Figure S5b.
Response:
Thanks for your comment. The mistake is corrected in the revised version and highlighted in red color.
Comment 5:
Page 9 line 256: Was this heating of the sample done at the same heating rate as that in the DSC thermogram (Fig. 6a)? If yes, then why is the material still crystalline? Shouldn’t it be amorphous according to the author’s discussion in Section 2.3 (line 229)?
Response:
The heating rate of the sample was done at the same heating rate as that in the DSC thermogram (10°C·min-1). Unfortunately, for technical reasons, it had been impossible to do VT-PXRD analysis to prove our assumption about sample amorphization. Instead, the PXRD pattern of the dehydrated sample was recorded after the sample was cooled to room temperature (Fig. S5b). That’s why fractional crystallization of the analyzed powder could have occurred. However, it was found that sample crystallinity after dehydration was significantly lower than crystallinity of the initial sample, which proved by the intensity reduction of the peaks by 2-5 times. Therefore, when the sample was reheated, there was no endotherm on the DSC curve, indicating the melting of the crystalline sample.